# COVID-19 diagnosis from CT scans and chest X-ray images using low-cost Raspberry Pi

**Khalid M. Hosny**[1]*, **Mohamed M. Darwish**[2], **Kenli Li**[3]*, **Ahmad Salah**[1,3]

**1** Faculty of Computers and Informatics, Zagazig University, Zagazig, Egypt, **2** Faculty of Computers and Informatics, Assiut University, Assiut, Egypt, **3** College of Computer Science and Electrical Engineering, Hunan University, Changsha, China

* k_hosny@yahoo.com (KMH); lkl@hnu.edu.cn (KL)

## Abstract

The diagnosis of COVID-19 is of vital demand. Several studies have been conducted to decide whether the chest X-ray and computed tomography (CT) scans of patients indicate COVID-19. While these efforts resulted in successful classification systems, the design of a portable and cost-effective COVID-19 diagnosis system has not been addressed yet. The memory requirements of the current state-of-the-art COVID-19 diagnosis systems are not suitable for embedded systems due to the required large memory size of these systems (e.g., hundreds of megabytes). Thus, the current work is motivated to design a similar system with minimal memory requirements. In this paper, we propose a diagnosis system using a Raspberry Pi Linux embedded system. First, local features are extracted using local binary pattern (LBP) algorithm. Second, the global features are extracted from the chest X-ray or CT scans using multi-channel fractional-order Legendre-Fourier moments (MFrLFMs). Finally, the most significant features (local and global) are selected. The proposed system steps are integrated to fit the low computational and memory capacities of the embedded system. The proposed method has the smallest computational and memory resources,less than the state-of-the-art methods by two to three orders of magnitude, among existing state-of-the-art deep learning (DL)-based methods.

**Data Availability Statement:** The data underlying this study are available on Kaggle (https://www.kaggle.com/plameneduardo/sarscov2-ctscan-dataset).

## 1 Introduction

COVID-19 pandemic affects the lifestyle of the entire world. New challenges are raised for human beings to use the existing knowledge to fight COVID-19 disease. One of these challenges is COVID-19 disease diagnosis using images of chest X-ray [1]. COVID-19 chest radiographs outline bilateral air-space consolidation, as described in the disease characteristics [2].

As DL-based methods are successfully utilized to solve different problems [3, 4], there are several attempts to use chest X-ray and CT scan images to detect COVID-19 cases [5, 6]. For instance, Apostolopoulos and Mpesiana [7] utilized a DL model to classify the X-ray images of patients into one of three classes: bacterial pneumonia, COVID-19 disease, and normal cases. Apostolopoulos and his coauthor used the deep transfer learning approach with four

**Funding:** Kenli Li was supported in part by the National Natural Science Foundation of China under Grant 61702170.

**Competing interests:** The author(s) received no specific funding for this work.

architectures, namely, VGG-19 [8], MobileNet [9], Inception [10], and Xception [11]. Their proposed method has the highest accuracy (98.75%) using the VGG-19 model.

Generally, the DL-based classification methods achieve the highest reported accuracy rates. Despite their classification high accuracy rates, running the deep learning models require very expensive computational resources with high specifications. This high-cost processing process might be affordable for large hospital in first-world countries, but hospitals in developing countries and rural areas do not have such expensive computational resources. To reduce the computational cost, Howard et al. [9] build a deep learning model that consumes fewer resources while sacrificing the accuracy rates. Despite the trial of Howard and his colleagues, successful DL-based classification models still require extremely expensive computational machines with high configuration.

Recently, orthogonal moments were utilized to extract features form color images and successfully used in various applications such as recognition of bacterial species [12]. Since medical images have fine details; thus, the task of extracting these features requires highly accurate descriptors. The recent fractional-order descriptors [13] enable the proposed system to extract high-accurate global features from the input CT scan or X-ray images.

These fractional-order descriptors have many characteristics as follows:

1. Their orthogonality enables the representation of medical images without information redundancy.

2. These descriptors are invariant with rotated, scaled and translated images, which improves the classification rates.

3. There is significant robustness against common noise, such as speckle.

4. The MFrLFM descriptors have much faster computation times than other moments.

The computational challenges of DL-based methods and the success of orthogonal moments in classification problems motivate the authors to develop a cost-effective diagnosis system of COVID-19 cases (i.e., less than 100 USD), which classifies input chest scan images such as X-ray and CT into COVID-19 or other lung disease with high classification accuracy rates. The main contributions of this work are as follows.

1. The proposed work is the first system to utilize the Linux embedded system to diagnose COVID-19 cases from the CT scan or X-ray images to the best of the authors' knowledge. Besides, the proposed system can run on any embedded system that supports running the Python code. The proposed system consists of two separate classifiers, one used for classifying the chest X-ray images and another model to classify the CT scan images.

2. The proposed system is designed to be a memory-efficient classification model, as state-of-the-art methods are DL-based methods with huge memory requirements. Thus, the proposed classifier model's main impact is that it becomes possible to obtain a high accuracy rates under a limited memory condition for predicting COVID-19 cases from chest CT and X-ray images.

3. The proposed system is the first system to utilize MFrLFMs moments for global features extraction from chest CT scans or X-ray images. Besides, the Local Binary Pattern (LBP) is utilized for extracting the local features of the input images.

The remainder of the paper is organized as follows. Section 2 exposes the required background of the used techniques and platform. Section 3 discusses the related work. Section 4

describes the proposed system. Section 5 evaluates the proposed system performance. Finally, the work is concluded in Section 6.

## 2 Preliminaries

### 2.1 Local binary patterns

The LBPs algorithm has two advantages: robustness to monotonic grayscale changes and low computational cost [14]. The effective performance of the LBPs operator is thoroughly discussed in [15]. LBPs have been utilized in various application domains including medical images classification, texture classification, and facial micro-expression recognition [16].

The LBP algorithm's basic idea is to assign a certain value, called a code, to each pixel. This pixel's value (i.e., code) encodes the local features of the $3 \times 3$ neighborhood window of the eight neighbor cells, as explained in [15]. In a $3 \times 3$ window, the value of the central pixel is considered the threshold. If the value of any neighbor pixel is less than the threshold, then this neighbor pixel is set to zero; otherwise, the neighbor pixel value is set to one. For example, the threshold of the $3 \times 3$ window, which is a portion of the image, in Fig 1(a), is 131. In Fig 1(b), each neighbor pixel is set to zero or one depending on the threshold value (i.e., 131). Then, the weight of each pixel is multiplied by the pixel value (i.e., zero or one). The LBP code/value, which is assigned to the window central pixel, is the summation of the multiplied value of all of the eight neighbors.

### 2.2 Multichannel fractional-order Legendre-Fourier moments

The RGB color image defined using the $f(r, \theta)$ intensity function is represented in three primary channels as $f(r, \theta) = (f_R(r, \theta), f_G(r, \theta), f_B(r, \theta))$ [17].

The MFrLFMs are:

$$FrM_{pq}(f_C) = (2p + 1)/2\pi \int_0^{2\pi} \int_0^1 f_C(r, \theta)[W_{pq}(r, \theta)]^* r dr d\theta \tag{1}$$

where $C$ denotes each primary channel (R-, G- or B-); $p$ and $q$ are the moment order and repetition, respectively; $|p| = 0, 1, 2, 3, \ldots\ldots\infty$, $|q| = 0, 1, 2, 3, \ldots\ldots\infty$.

The function is

$$W_{pq}(r, \theta) = L_p(\alpha, r)e^{(iq\theta)} \tag{2}$$

(a)

Sample window

| 5 | 221 | 88 |
|---|-----|-----|
| 151 | 131 | 91 |
| 132 | 41 | 144 |

(b)

LBP code

| 0 | 1 | 0 |
|---|---|---|
| 1 |   | 0 |
| 1 | 0 | 1 |

× 

| 1 | 2 | 4 |
|---|---|---|
| 128 |   | 8 |
| 64 | 32 | 16 |

= 2+16+64+128 =210

**Fig 1. An example of LBP code calculation of a single window.** ((a)) $3 \times 3$ Sample window, ((b)) The calculation of the LBP code of the input windows.

where the fractional-order Legendre polynomials $L_p(\alpha, r)$ are:

$$L_p(\alpha, r) = \sqrt{\alpha} \sum_{k=0}^{p} (-1)^{(p+k)} \frac{(p+k)! r^{\alpha k + \frac{(\alpha-2)}{2}}}{(p-k)!(k!)^2} \tag{3}$$

Because direct computation using Eq 3 is time-consuming, the three-term recurrence relation is utilized as an alternative.

Since,

$$|M_{(p,q)}(f_C^{rot})| = |M_{(p,q)} f_C|, C \in R, G, B \tag{4}$$

Eq 4 shows that the rotation does not affect the magnitude values of MFrLFMs. The MFrLFMs scale invariants forms are:

$$\varphi_{pq} = \sum_{k=0}^{p} \frac{2p+1}{2k+1} \left( \sum_{i=k}^{p} (FrM_{00}(f_C)^{(-(2i+3)/3)} C_{pi} d_{ik}) \right)$$
$$FrM_{kq}(f_C) \tag{5}$$

where coefficients $C_{pi}$ and $d_{ik}$ are [18]:

$$C_{pi} = (-1)^{(p+i)} \frac{(p+i)!}{(p-i)!(i!)^2} \tag{6}$$

$$d_{ik} = \frac{(2k+1)(i!)^2}{(i+k+1)!(i-k)!} \tag{7}$$

A highly accurate kernel-based computational framework [19] is used to compute MFrLFMs as follows:

$$FrM_{pq} = \frac{(2p+1)}{2\pi} \sum_{i} \sum_{j} I_p(r_i) J_q(\theta_{ij}) \hat{f}_c(r_i, \theta_{i,j}) \tag{8}$$

The interpolated function $\hat{f}_c(r_i, \theta_{i,j})$ is calculated from the intensity functions of the original image. This task can be achieved using the cubic interpolation, as explained in [20].

Based on Eq 8, the radial and polar kernels are:

$$J_q(\theta_{ij}) = \int_{V_{i,j}}^{V_{i,j+1}} e^{(-\hat{i}q\theta)} d\theta \tag{9}$$

$$I_P(r_i) = \int_{U_i}^{U_{i+1}} L_p(\alpha, r) r \, dr = \int_{U_i}^{U_{i+1}} R(r) dr \tag{10}$$

Eq 9 shows that kernel $J_q(\theta_{ij})$ is exactly evaluated as follows:

$$J_q(\theta_{i,j}) = \begin{cases} \frac{\hat{i}}{q}(e^{-\hat{i}q V_{(i,j+1)}} - e^{-\hat{i}q V_{i,j}}) & q \neq 0 \\ V_{i,j+1} - V_{i,j} & q = 0 \end{cases} \tag{11}$$

An accurate numerical integration approach is used for calculating the integration in Eq 11.

## 2.3 Raspberry Pi: A Linux embedded system

Raspberry Pi is a single-board computer or a Linux embedded system; it is an open-source ecosystem. It is a cost-effective, lightweight, and portable computer. Raspberry Pi has been utilized in several machine learning applications such as computer vision and image classification [21].

Because Raspberry Pi hardware supports Linux OS, we can benefit from the Python programming language and its powerful packages, especially the Scikit-learn package [22]. Thus, Raspberry Pi hardware can run many machine learning tasks. Another advantage of the Raspberry Pi model is that one of its versions has a multi-core CPU, which enables the acceleration of the running programs by providing parallel implementations of the utilized algorithms.

In [23], the authors discussed the methodology of task division on a Raspberry Pi hardware using OpenMP [24] and MPI [25]. Then, the authors utilized this parallel implementation over a cluster of Raspberry Pi devices to address the problem of edge detection. Another example of the parallel implementation of Raspberry Pi devices is reported in [26]. The authors proposed using a cluster of Raspberry Pi 2 to accelerate the 3D wavelet transform and make it portable.

A user can realize the overall performance of the Raspberry Pi 4 model B as the performance of an entry-level x86 PC, as shown in Fig 2. Raspberry Pi 4 model B utilizes a 64-bit CPU with four cores. The Raspberry Pi 4 model comes with three different options of main memory (i.e., RAM), namely, 1 GB, 2 GB, and 4 GB. For the display options, Raspberry Pi 4 Model B supports a dual-display option (i.e., two micro-HDMI ports). The quality of the display is as high as 4K video resolution.

In addition, Raspberry Pi 4 model B supports different connectivity methods: wireless connection via a dual-band 2.4/5.0 GHz wireless LAN port, Gigabit Ethernet, and Bluetooth 5.0. These features allow one to connect the Raspberry Pi model to any other device, which supports IoT applications. In addition, the USB 3.0, Raspberry Pi model has three USB ports: one port for power connection and two ports for attaching four different peripherals (e.g., mouse and keyboard).

## 3 Related work

There is much-conducted research that addressed COVID-19 diagnosis from chest CT scans or X-ray images with machine learning techniques. These efforts can be classified based on which deep architecture was utilized by the proposed work. In the following, we discuss representative research works based on the utilized deep architectures.

The DL-based models of COVID-19 diagnosis from chest CT scans or X-ray images are considered the mainstream. Several classification models are proposed, while the main difference is the utilized deep architecture (e.g., Residual Network (ResNet), VGG, Dense Convolutional Network (DenseNet), etc.).

Convolutional Neural Networks (CNNs) [27] is considered the most used deep architecture for image classification. In [28], the authors proposed a CNN-based model for detecting COVID-19 cases from chest X-ray images. They proposed two models; the first one is a binary classifier with two possible outcomes, COVID-19 and Non-COVID-19. The second proposed model is a multi-class classifier model with three possible outcomes, Pneumonia, COVID-19, and Non-COVID-19. Their proposed model classification accuracy rates are 98% and 87%, respectively. In [29], Abd Elaziz et al. utilized two classifier models and two different chest X-ray image datasets to detect COVID-19 cases. Then, they proposed several CNN-based methods for the purpose of comparison. The proposed model accuracy rates were 96% and 98% for

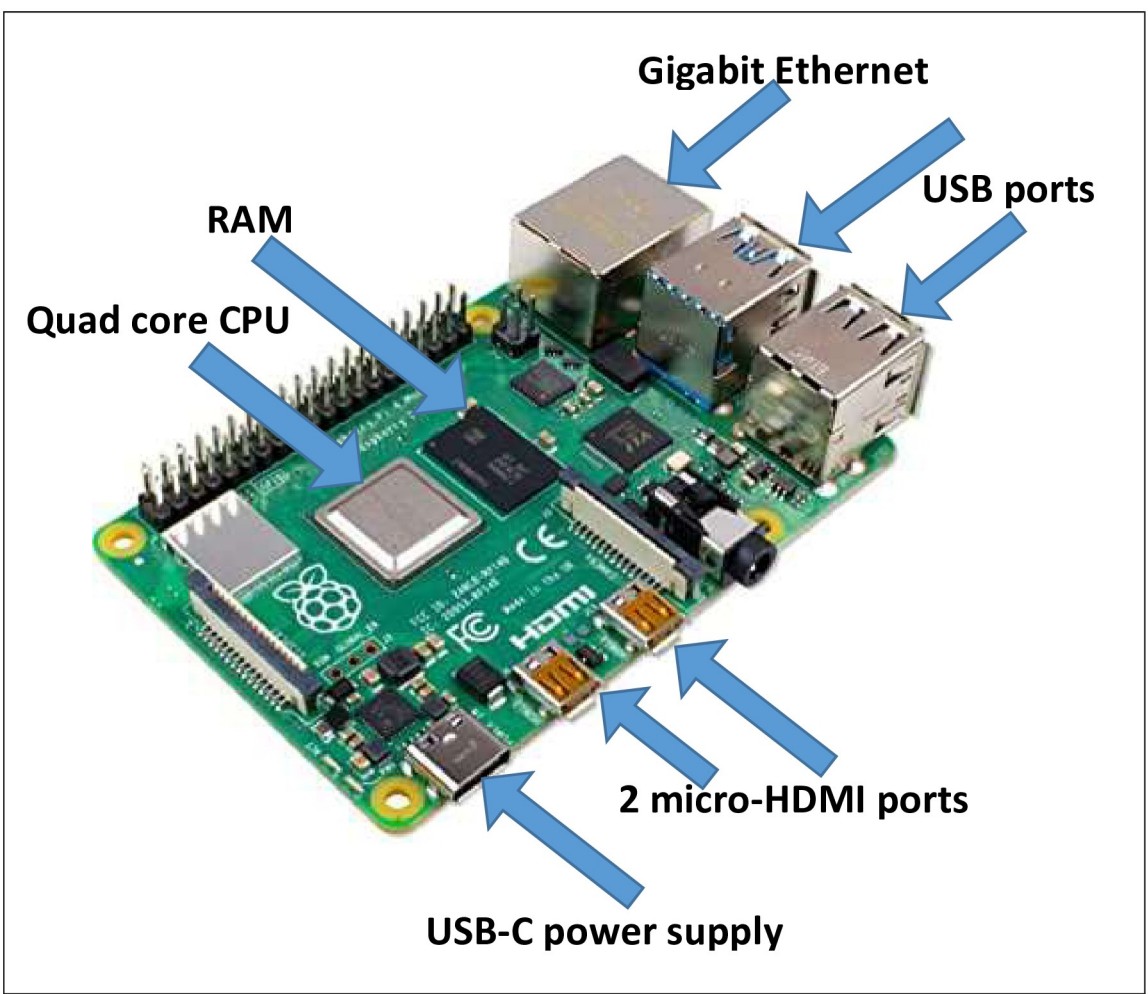

**Fig 2. Raspberry Pi 4 model B hardware design.**

the first and second datasets, respectively. In addition, the authors in [30] compared several CNN-based COVID-19 detection models.

The ResNet architecture [31] has a significant performance in image classification on several image datasets. In [32], the authors utilized the deep transfer learning approach to train a ResNet architecture for the sake of automatic COVID-19 detection from chest X-ray images. The authors utilized a dataset of 350 normal, 350 Pneumonia, and 210 COVID-19 chest X-ray images. The classification accuracy rate is 94.28%.

The DenseNet architecture is proposed in [33]. In DenseNet, a layer receives inputs from all previous layers; meanwhile, the same layer passes on its feature-maps to all of the following layers. As CT scan images play a vital role in COVID-19 cases automatic detection, several works utilized CT scan images [34–36]. The authors in [37] proposed using deep transfer learning on DenseNet-201 architecture to classify the suspected case as COVID-19 or normal using the patient's CT scan image. They trained the proposed classifier model using a dataset consisting of 2,492 CT scans. The achieved classification accuracy rate is 96%. In [38], the authors proposed using a portable on-device system to detect COVID-19 patients based on the chest X-ray images automatically. The proposed system can follow-up on the case progression as well. The authors utilized the DenseNet-121 architecture with the help of deep transfer

learning to build the classifier model. The highest reported classification accuracy by the proposed system is 88%.

In [39], the authors proposed a 3D deep CNN-based model to recognize COVID-19 cases using CT volumes automatically. The authors proposed generating 3D lung masks using the pre-trained UNet [40], and then these generated masks are classified. The obtained classification accuracy is 90%. In the same context, several research works proposed different tasks on COVID-19 CT scan images. For instance, the authors in [41, 42] proposed two segmentation methods for removing the noise data from the input image, as a pre-processing step for the classification task. These proposed segmentation methods eased the classification task and resulted in improving the classification accuracy rates.

The VGG deep architecture achieves high classification accuracy rates despite its huge memory requirements. In [43], the authors utilized a dataset of 592 CT scan images with two classes COVID-19 and normal. Then, they proposed the CTnet-10 model, a binary classifier model. This proposed model's classification accuracy rate is 82.1% while utilizing the pre-trained VGG-19 model for the classification task yields an accuracy of 94.5%. Another VGG-based model is proposed in [44]. The authors proposed a multi-class classifier to classify a chest X-ray image as COVID-19, pneumonia, or normal. The utilized dataset consists of 360 images. They proposed creating feature maps from the X-ray images, and then the vectorized version of these feature maps are classified using the VGG-16 architecture. They utilized the deep transfer learning approach by using the saved VGG-16 weights as trained on the ImageNet dataset. Besides, they proposed adding an output layer for the three possible classification outcomes. The classification accuracy rate is 91%.

## 4 Proposed system

The proposed system consists of four main phases, as shown in Fig 3. The first phase includes extracting the local features using the LBPs algorithm. The second phase includes global features using MFrLFMs. In the third phase, the local and global features are combined and then a feature selection method is applied to select the most significant features. Finally, the fourth phase includes a binary classifier that takes the selected local and global features as an input to classify the input image as COVID-19 disease or other diseases.

### 4.1 Local features using the LBPs

The LBP feature vector is computed as a $1 \times N$ vector, where $N$ is the number of extracted local features. The LBPs algorithm partitions the input image into non-verlapping windows. A wider window size corresponds to less computational complexity and fewer details of the collected local features. In the proposed system, the number of neighbors $P$ is set to 8. Thus, the total number of extracted local features is $N = (P \times P - 1) + 3 = (8 \times 7) + 3 = 59$ features.

### 4.2 Global feature extraction using MFrLFMs

The sequential computations of MFrLFMs are inconvenient for multicore CPU without loop fusion, since the computations consist of four nested loops. Thus, we utilized the loop fusion technique to the outermost two loops to parallelize the sequential computations of MFrLFMs. Thus, the iterations of this fused loop can be independently computed.

Since Raspberry Pi has at most four cores, the loop fusion of the outermost two loops provides a sufficient number of independent iterations. The iteration number of the two fused loops of MFrLFM computation is mapped to the original loop iteration numbers as shown in

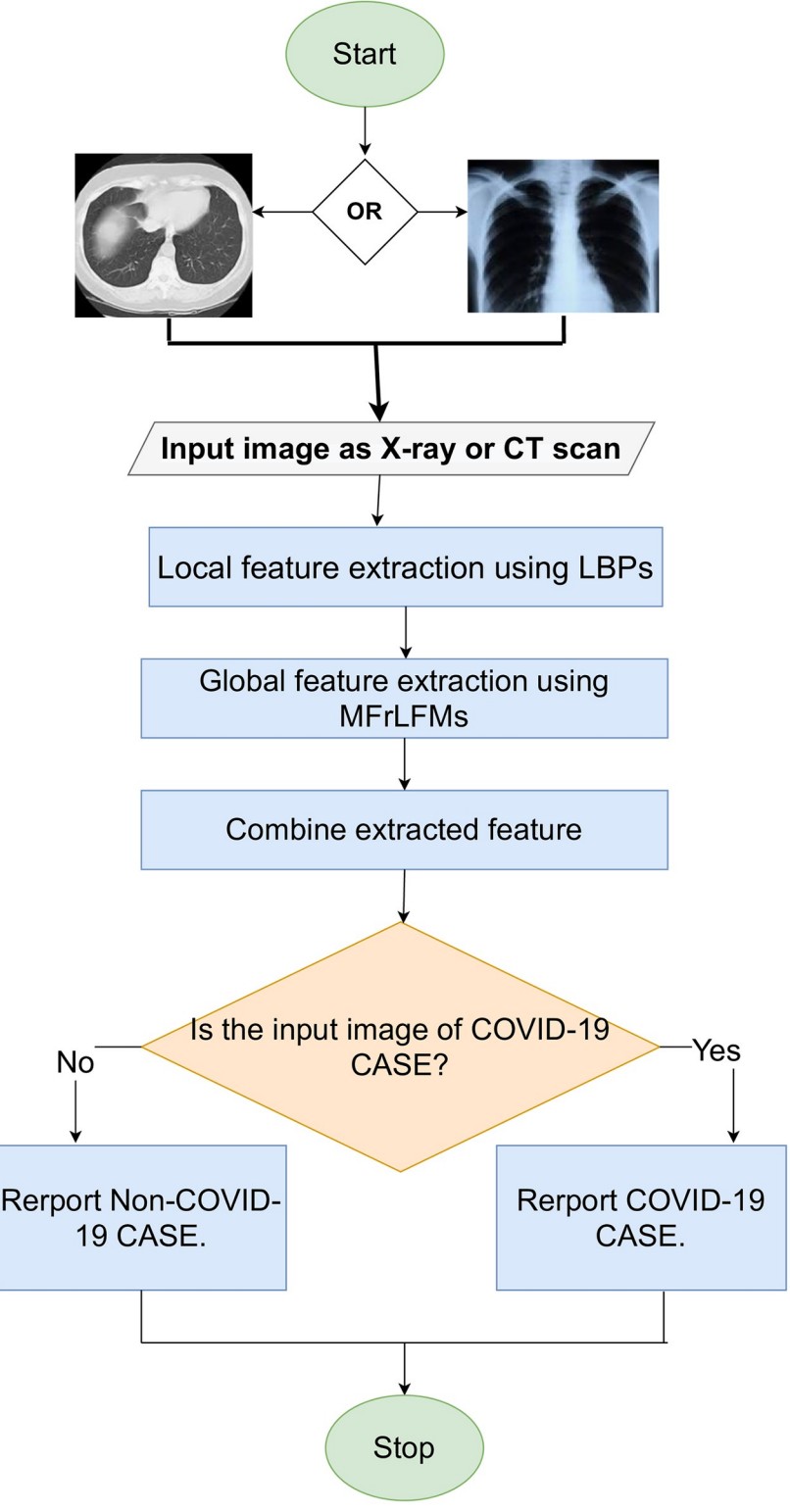

**Fig 3. Flowchart of x-ray image and CT scan classification models.**

Eq 12 for two loops.

$$
\begin{aligned}
p &= i_{fused}/(qmax + 1) \\
q &= i_{fused} \bmod (qmax + 1)
\end{aligned}
\tag{12}
$$

where $i_{fused}$ is the iteration number of the four fused loops, $i_{fused} \in [0, (pmax + 1) \times (qmax + 1)]$ for a two-loop fusion.

Algorithm 1 lists the parallel implementation of the MFrLFM computations. In Algorithm 1 line 1, the algorithm divides the iterations of the outer loop over only $p$ parallel resources, i.e., Raspberry Pi CPU cores. This task can easily be accomplished using the OpenMP directive *#pragma omp parallel for num_threads(p)*. This OpenMP evenly divides $(pmax + 1) \times (qmax + 1)$ iterations over the available $p$ threads/cores.

In Algorithm 1 line 2, the **for** loop represents two fused loops of kernels $p$ and $q$. Iterator $i_{fused}$ goes through $(pmax + 1) \times (qmax + 1)$ iterations; each iteration represents unique $p$ and $q$ values. Thus, the variable $i_{fused}$ should be mapped to the corresponding $p$ and $q$ values, as listed in lines 4 and 5, using Eq 12. Line 6 resets the accumulative variables of each $M_{p,q}$ moment. The **for** loop in Line 7 goes through all of the $M$ image rings. Similarly, the **for** loop in Line 8 goes through each sector $r$. Line 9 computes the kernel value. To compute the kernel value, two terms should be multiplied; the radial kernel value is accessed using the $p$ and $ring$ values, and the repeating kernel is accessed using the $ring$, $sec$, and $q$ values. Lines 10-12 compute the moment of the three channels, i.e., red, green, and blue. At each of these three lines, the image pixel is accessed by the term $r\_image[ring][sec]$ using the $ring$ and $sec$ values and multiplied by the kernel value, as computed in Line 9. Finally, the $M_{p,q}$ moment is computed by multiplying the value computed within the loop by a constant.

Algorithm 1 consists of three nested loops. The time complexity of the first loop is $O\left(\frac{pmax \times qmax}{p}\right)$. The second and third loops iterate over each pixel of the $N \times N$ pixels of the input image. The time complexity of the second and third inner loops is $O(N^2)$. Thus, the time complexity of Algorithm 1 is the multiplication of time complexity of these three loops. Using $p$ parallel resources, the time complexity of Algorithm 1 (i.e., MFrLFMs) is $O\left(\frac{pmax \times qmax \times N^2}{p}\right)$.

The time complexity of computing the LBP algorithm is $N^2$. There are $N^2$ pixels per the input image; for each pixel, a binary patter of size eight is generated, where each neighbor contributes by one bit. Thus, the time complexity of computing the $N^2$ LBP codes is $O(8 \times N^2) = O(N^2)$. As the LBPs can be calculated independently, the LBPs algorithms can easily be parallelized by dividing the $N^2$ LBP codes computation over $p$ parallel resources. Thus, the final time complexity of the local feature extraction phase is $O(N^2/p)$.

The proposed systems' overall time complexity equals the time complexity of the summation of local and global feature extraction time complexities. Thus, the proposed system overall time complexity is $O\left(\frac{pmax \times qmax \times N^2}{p} + \frac{N^2}{p}\right) = O\left(\frac{pmax \times qmax \times N^2}{p}\right)$.

The space complexity of the MFrLFMs algorithm is $O(pmax \times qmax)$, as the algorithm stores the computed moments in a 2D matrix of $pmax$ rows and $qmax$ columns regardless the image size, i.e., the value of $N$. On the other hand, the space complexity of the local feature extraction phase by the LBP algorithm is $O(N^2)$. This is because the LBPs algorithm stores a code for each pixel of the $N^2$ image pixels. Thus, the overall space complexity of the proposed method is $O(N^2 + pmax \times qmax) = O(N^2)$, as $pmax \ll N$ and $qmax \ll N$.

**Algorithm 1** The parallel algorithm of MFrLFMs computations.

```
1 Divide the following for iterations over p cores
2 for i_fused = 0:
3 (pmax + 1) × (qmax + 1) do
```

```
 4 p = i_fused /(qmax + 1)
 5 q = i_fused mod (qmax + 1)
 6 r = g = b = 0
 7 for ring = 1: M do
   8 for sec = 1: S × (2 × i + 1) do
     9 kernel_val = I_p[p][ring] × Iq[ring][q][sec]
     10 r += r_image[ring][sec] × kernel_val
     11 g += g_image[ring][sec] × kernel_val
     12 b += b_image[ring][sec] × kernel_val
   end for
 end for
 13 Red_M_{p,q}[p][q] = r × ((2 × p) + 1)/(2 × PI)
 14 Green_M_{p,q}[p][q] = g × ((2 × p) + 1)/(2 × PI)
 15 Blue_M_{p,q}[p][q] = b × ((2 × p) + 1)/(2 × PI)
 end for
```

## 4.3 Feature selection and classification

The last step to prepare the data for the classification task is to select the most significant features to classify the images. The number of extracted local features of each input image is 59, as discussed in 4.1. The number of extracted global features of each input image is $(pmax + 1) \times (qmax + 1)$ features. For example, if $pmax = qmax = 30$, then each image is represented by 961 global features. Thus, the total number of local and global features for an image when $pmax = qmax = 30$ is 1,020 features. In other words, each image is represented by 1,020 decimal values.

We proposed applying a feature selection technique to remove any irrelevant, redundant, and noise features. The feature selection reduces the classifier training time and classifier prediction time, since extracting fewer features reduces the feature extraction phase time. To achieve this goal, we proposed using Sequential Feature Selector (SFS) greedy search technique. This greedy approach has $k$ iterations. The SFS method adds one feature at each iteration, which is the most significant feature; finally, The SFS algorithm selects the most $k$ significant features. If the SFS algorithm finds a subset of features with fewer than k features, this feature subset is reported. Thus, the number of selected features can be smaller than or equal to $k$. The SFS is executed one time; thus, it has no effect on the proposed method runtime nor the proposed method time complexity.

Once we selected the most significant $k$ features for all input images using the SFS technique, the dataset is ready to train the classifier. The proposed method is applicable for any binary classifier, COVID-19 or non-COVID-19 input image, where the input image can be a chest X-ray or a CT scan image. We proposed two separate classifier models for each of the two image types.

## 5 Experimental results

### 5.1 Dataset

The utilized dataset consists of eight lung diseases from eight chest X-ray dataset [45]: 1) Atelectasis; 2) Cardiomegaly; 3) Effusion; 4) Infiltration; 5) Mass; 6) Nodule; 7) Pneumonia; 8) Pneumothorax. Each lung disease has 212 images. These images of the eight lung diseases are collected in one image class, which is called Non-COVID-19 diseases. The second class consists of 212 images of chest X-ray of COVID-19 patients [46]. This data collection approach results in unbalanced classes, as the first class has $212 \times 8 = 1,696$ images and the second class has 212 images. In addition, we used a dataset of CT scans of COVID-19 patients and other lung diseases. The second dataset consists of 2,842 images classified in two classes: COVID-19

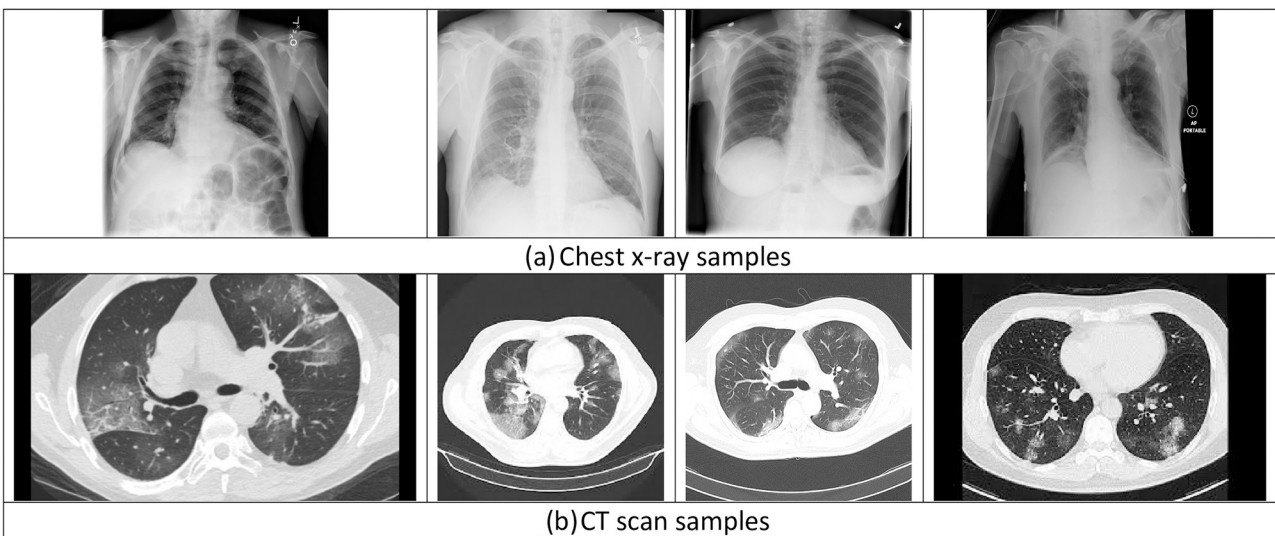

**Fig 4. Sample of the two datasets: (a) chest X-ray images [45]; (b) CT scans [47].**

class with 1,252 images and non-COVID-19 class with 1,230 images [47]. Fig 4 shows samples of these two datasets.

The utilized datasets are split into 80% and 15% as training and test sets, respectively. These two sets are used to train and test the proposed models and the comparison method. Besides, the cross-validation technique was utilized, where the number of folds was five. In other words, the dataset was divided into five different ways. Finally, the hyperparameters of these methods were set to the default values during the training phase for the methods of comparison.

## 5.2 Setup

Experiments are performed on a Raspberry Pi 4 Model B with 4-cores CPU. The utilized OS is 64-bit Linux. The implementations were written in the C++ and the Python programming languages. C++ is used to implement the local- and global-feature extraction algorithms, and Python is used to implement the image classifier. We used the standard OpenMP thread library [24] for CPU multi-core implementation. The reported results are the average of running each experiment three times. Fig 5 shows the result of the proposed system on Raspberry Pi 4 Model B, where the input image is classified as a chest X-ray of a COVID-19 patient.

The MFrLFM radial and repeating kernel order is set to 31. Thus, the number of global features is 961. The number of local features is 59. The total number of utilized features is 1,020. After the SFS feature selection method has been applied, the chest X-ray image classifier and CT scan classifier are trained on 26 global plus 15 local features (i.e., 41 features).

## 5.3 Results

Table 1 lists three different accuracy metrics to evaluate the proposed methods, including the accuracy, AUC, and F1-score. The proposed method achieved comparable results with comparison to other deep learning methods in terms of accuracy, AUC, and F1-score metrics. Table 2 lists the required memory and prediction time of the proposed trained classifier. Table 3 lists memory requirements of the proposed method and state-of-the-art models. As

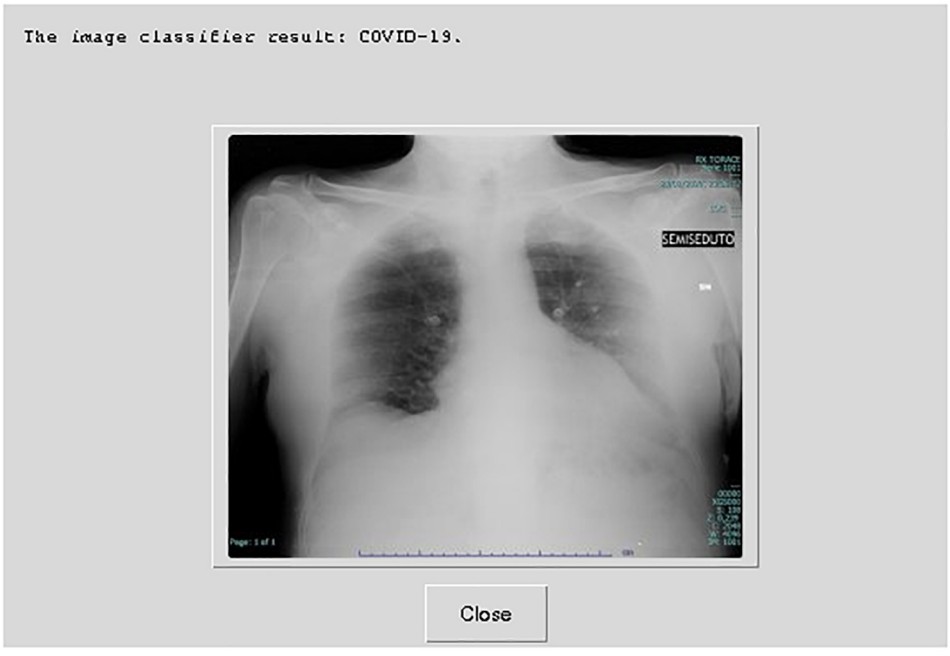

**Fig 5. Proposed system on Raspberry Pi 4 Model B.**

listed in Table 3, the proposed system has the least memory requirements and prediction time in comparison to the other deep-learning-based methods.

To evaluate the ROC AUC values, Fig 6 depicts the receiver operating characteristic (ROC) curve, and Fig 7 depicts the precision curve of the proposed method. Figs 6 and 7 show the efficiency of the proposed system to classify the input chest X-ray CT scan as a COVID-19 disease or other disease.

**Table 1. Accuracy of the proposed models.**

| data | Accuracy(%) | F1-score | AUC |
|---|---|---|---|
| chest X-ray | 99.3±0.2% | 93.1±0.2% | 94.9±0.1% |
| CT scans | 93.2±0.3% | 92.1±0.3% | 93.2±0.3% |

**Table 2. Required memory and the prediction time in seconds of the two proposed models on Raspberry Pi.**

| data | Memory | Prediction time (s) |
|---|---|---|
| chest X-ray | 3MB | 10 |
| CT scans | 3MB | 10 |

**Table 3. The required memory of the propped model and state-of-the-art models on Raspberry Pi.**

| Method | Memory |
|---|---|
| The proposed system | 1MB |
| VGG19 [8] | 1,406MB |
| MobileNet v2 [9] | 13MB |
| Inception-3 [10] | 232MB |
| ResNet-50 [48] | 101MB |

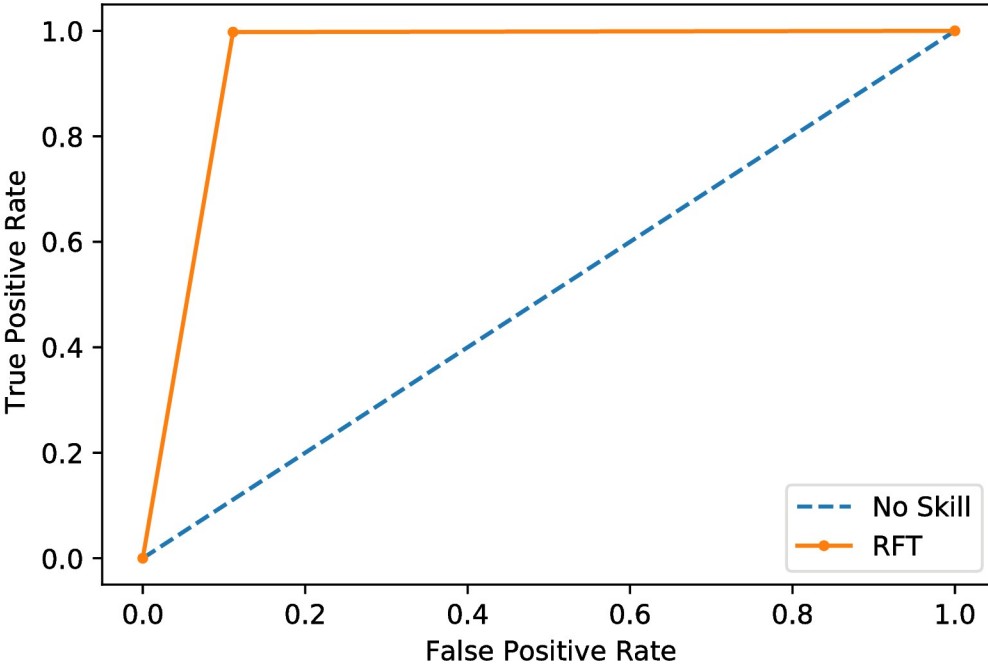

**Fig 6. ROC curve of the proposed X-ray image classifier model.**

Finally, the precision, recall, and confusion matrices are examined for the two proposed models. Figs 8 and 9 show the precision-recall scores of the proposed two models. Besides, the confusion matrices are depicted in Figs 10 and 11 for the two proposed classifiers.

The results above outline the memory requirements gap between of the proposed classifiers and state-of-the-art methods; the proposed models require memory spaces less the existing

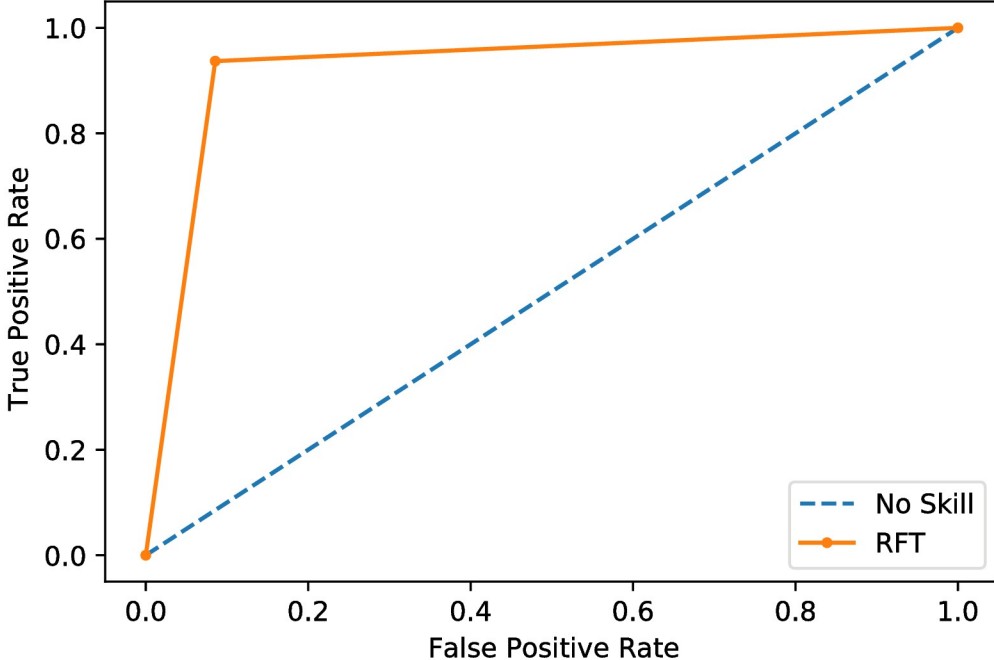

**Fig 7. ROC curve of the proposed CT scan classifier model.**

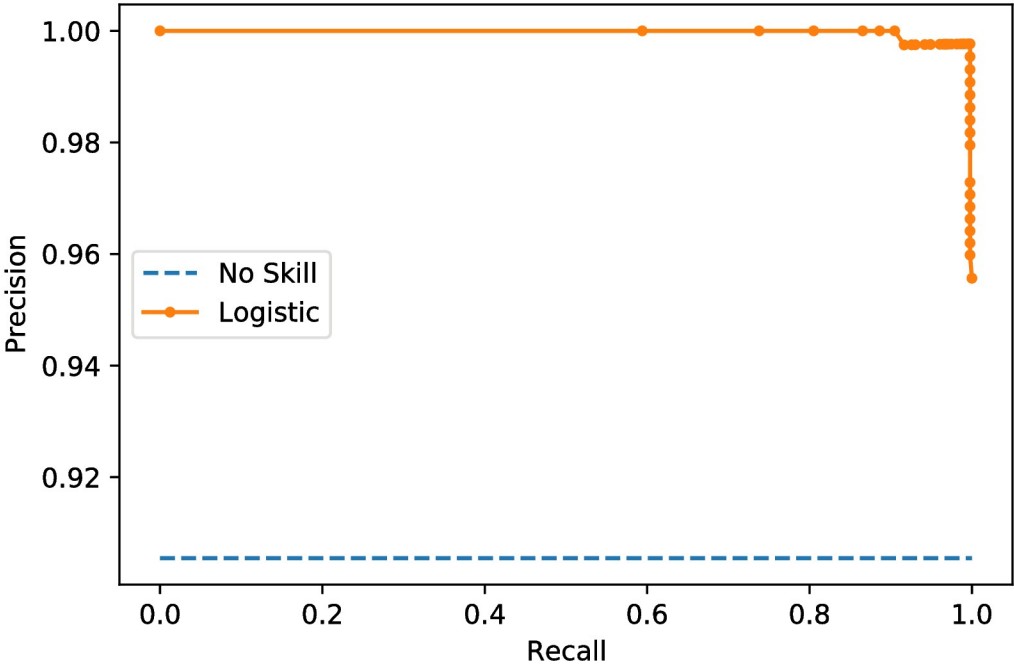

**Fig 8. Precision-recall score curve of the proposed X-ray image classifier model.**

methods by 2-3 orders of magnitude, as listed in Table 3. Thus, the main goal of this research is achieved, as a small-sized (i.e., 3 MB) model for COVID-19 diagnosis can fit the low-memory embedded system. In addition, the proposed models maintain the accuracy rates of state-of-the-art methods.

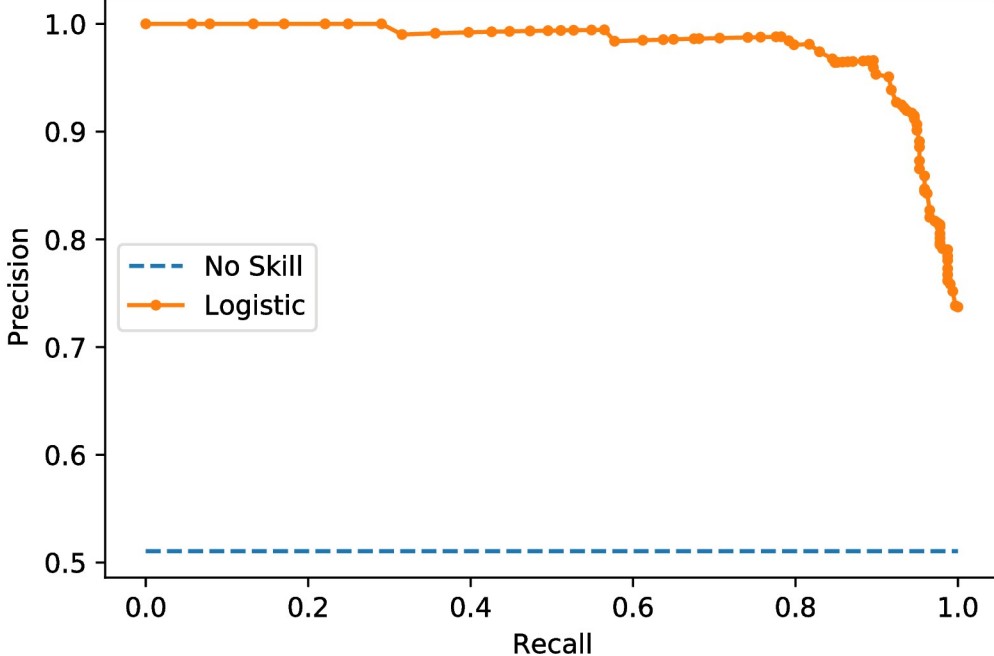

**Fig 9. Precision-recall score curve of the proposed CT scan classifier model.**

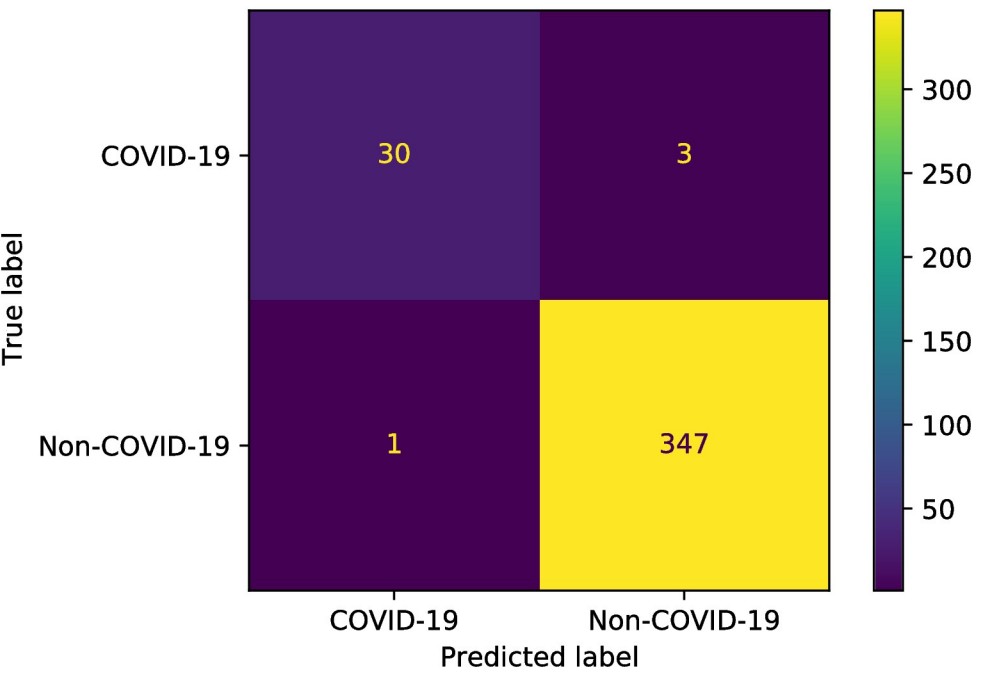

**Fig 10. Confusion matrix of the proposed X-ray classifier model.**

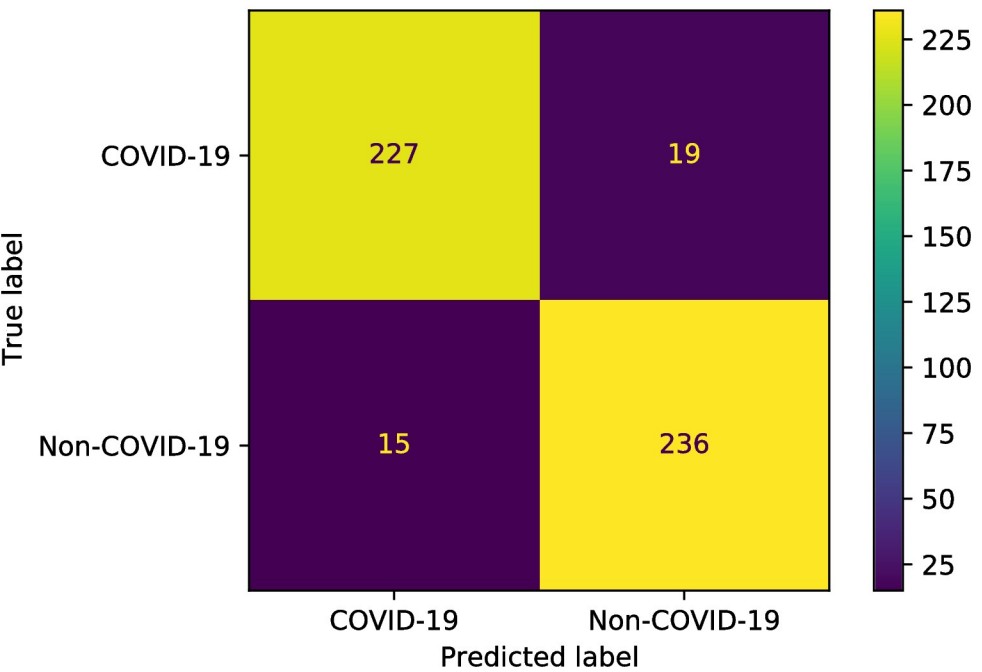

**Fig 11. Confusion matrix of the proposed CT scan images classifier model.**

## 6 Conclusion

In this work, we proposed two low-cost image classifier models that can operate on a Linux-embedded system, i.e., Raspberry Pi, to automatically detect COVID-19 cases on two types of imagery data, namely, chest X-ray and CT scan images. To our knowledge, this is the first time to achieve this task. The proposed system consists of several steps. First, the proposed methods extract the local features using LBP and then extract the global features using MFrLFMs moments from the input image. Second, the combined local and global features represent the input chest X-ray or CT scan image's final features. Finally, a classifier is trained to distinguish COVID-19 cases from the chest X-ray or CT scan images of other lung diseases. The proposed classification models require the smallest amount of memory (approximately 3 MB), which makes these models suitable for computationally limited hardware. The two proposed models are evaluated on a chest X-ray dataset of 1,926 images and a CT scan dataset of 2,482 images; each dataset has two classes (i.e., COVID-19 and other lung diseases). The proposed system has comparable scores on the evaluation metrics with state-of-the-art methods. At the same time, their computational and memory requirements are less than those of state-of-the-art DL-based methods by 2-3 orders of magnitude. As future work, the proposed system can be extended to classify more lung diseases. This can be achieved by proposing a multi-class classifier and utilizing the proper dataset.

## Author Contributions

**Conceptualization:** Khalid M. Hosny, Kenli Li, Ahmad Salah.

**Data curation:** Khalid M. Hosny.

**Investigation:** Khalid M. Hosny, Mohamed M. Darwish, Kenli Li.

**Methodology:** Khalid M. Hosny, Ahmad Salah.

**Resources:** Khalid M. Hosny, Mohamed M. Darwish, Ahmad Salah.

**Supervision:** Khalid M. Hosny, Kenli Li, Ahmad Salah.

**Visualization:** Khalid M. Hosny, Mohamed M. Darwish.

**Writing – original draft:** Khalid M. Hosny, Ahmad Salah.

**Writing – review & editing:** Khalid M. Hosny, Mohamed M. Darwish.

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
