## [Decision Letter · Decision Letter 0]

1 Mar 2021

PONE-D-21-04444

COVID-19 Diagnosis from CT Scans and Chest X-ray Images using Low-cost Raspberry Pi

PLOS ONE

Dear Dr. Hosny,

Thank you for submitting your manuscript to PLOS ONE. After careful consideration, we feel that it has merit but does not fully meet PLOS ONE’s publication criteria as it currently stands. Therefore, we invite you to submit a revised version of the manuscript that addresses the points raised during the review process.

I found this manuscript well written and interesting. As you will infer from below that there was a disagreement among the reviewers regarding enthusiasm for this work. Reviewer 1 was of the view that manuscript partly describes a technically sound piece of scientific research and recommended major revision. However,  Reviewer 2 and Reviewer 3 made certain suggestions for improvement and were of the view that your work describe technically sound piece of scientific research and recommended minor revision. 

After thorough consideration of comments of reviewers, my decision is "minor revision". Please incorporate comments raised by both reviewers. 

We look forward to receiving your revised manuscript.

Kind regards,

Gulistan Raja

Academic Editor

PLOS ONE

Journal Requirements:

"Kenli Li was supported in part by the National Natural Science Foundation of China under Grant 61702170."

Reviewers' comments:

Reviewer's Responses to Questions

**Comments to the Author**

1. Is the manuscript technically sound, and do the data support the conclusions?

Reviewer #1: Partly

Reviewer #2: Yes

Reviewer #3: Yes

2. Has the statistical analysis been performed appropriately and rigorously? 

Reviewer #1: No

Reviewer #2: Yes

Reviewer #3: Yes

3. Have the authors made all data underlying the findings in their manuscript fully available?

Reviewer #1: Yes

Reviewer #2: Yes

Reviewer #3: No

4. Is the manuscript presented in an intelligible fashion and written in standard English?

Reviewer #1: No

Reviewer #2: Yes

Reviewer #3: Yes

5. Review Comments to the Author

Reviewer #1: As far as one can see, the experimental work seems to have been carried out properly and the results are carefully presented, both in tables and in graphs, to compare this and earlier methods. However, the paper has following concern:

1. I dont understand exactely, why authors are fousing on Linux-embedded system, i.e., Raspberry Pi, to visually detect COVID-19 in chest X-ray and CT scans. Irrespective of platform, ultimately we can think of prediction accuracy with platform independent for better prospective.

2. Can the authors show the impacts of various proposed features to establish the proposal?

3. The training conditions when comparing with other methods should be discussed more specifically. The paper lacks this part.

4. Do the authors take different combinations between training and testing samples in experimentations to show the merit of the proposed method? Proper justification should be added for better clarity.

5. The title of the paper is very specific. I'll suggest modifying it to make it more generic using the term "directional codes".

6. The title of the paper is very specific. I'll suggest modifying it to make it more generic using the term "directional codes".

7. Whilst the English is generally quite good, there are quite a few minor grammatical errors, and a careful read through is needed to eliminate these.

8. More experimental analysis need to be done for better claity to the end reader.

9. The objective and motivation need to address properly in abstract section. Current form of abstract will not accept by research community.

10. The major contribution of the paper should be heighlited immediately after literature kind review.

Hope these above points will be more beneficial to the author for improvement of the paper.

Reviewer #2: This paper design a Raspberry Pi Linux embedded system for COVID-19 diagnosis from CT Scans and Chest X-ray Images. The cost of this system is the smallest among the deep learning-based models. My major concerns are list below:

-In the introduction section, should discuss some recent state-of-the-art models so that the readers can know more about the new techniques in this field. There are some literature that may be useful to further improve the quality of this section.

[ref1]Review of artificial intelligence techniques in imaging data acquisition, segmentation and diagnosis for covid-19, IEEE reviews in biomedical engineering, 2020;

[ref2]Inf-Net: Automatic COVID-19 Lung Infection Segmentation from CT Images， IEEE transactions on medical imaging, 2020;

[ref3]Joint prediction and time estimation of COVID-19 developing severe symptoms using chest CT scan, MIA, 2021;

[ref3]Severity assessment of COVID-19 using CT image features and laboratory indices, Physics in Medicine & Biology, 2020;

[ref4]Adaptive feature selection guided deep forest for covid-19 classification with chest ct, JBHI, 2020;

[ref5]JCS: An Explainable COVID-19 Diagnosis System by Joint Classification and Segmentation, TIP, 2021;

-As seen in Fig.3, the proposed system belongs to the multi-modality system, thus, the authors should discuss more these works.

Reviewer #3: This work presents a novel approach for rapid, on-device COVID-19 detection using Raspberry Pi. Despite the plethora of works on this topic, this one clearly stands-out due to the low computational requirements and the Raspberry Pi deployment. The experimental section is a bit short, but the results are convincing. I recommend acceptance, although a few things should be addressed:

Main points:

- Is this the first scientific report of using Raspberry Pi to diagnose COVID from medical images? If yes, please state so and if not, please cite relevant work. I quickly searched but couldn’t find anything very similar. I also recommend discussing the work by [2] since it also deals with on-device inference.

- Can you please comment on the overall time and space complexity?The elaboration about complexity in MFrLFMs is great (although there, at least a comment on space complexity would also be useful). It would be great to see a similar elaboration on the LBP method? I think if the entire pipeline (Fig3) can be expressed in terms of computational complexity in O-notation, this could be a major finding and contribution that might even be worth mentioning in the abstract. Also the time complexity could be briefly compared with a standard MLP/CNN to solidify how important this contribution is.

- You performed a binary classification. Although it’s questionable how practically relevant this is, it’s beyond the scope of this work. But could you please briefly comment whether/how this could be extended to a multi-class classification.

- Table 1: I understand that these results are based on three repeated runs. Could you please at least show the performance with one more digit precision and also indicate the standard deviation. Ideally, a cross validation should be performed to strengthen this finding.

- Please have a look at what “data availability” for this journal means. I understand the imaging data is public, but the requirements write e.g.: “For example, in addition to summary statistics, the data points behind means, medians and variance measures should be available.” I therefore had to click “No” at one of the questions.

Line 7 - 32: Correct me if I’m wrong, but this seems to be a random excerpt of the myriad of publications about COVID-19 detection with DL. I think these lines can be removed and replaced by a few, generic sentences about the efforts in the field. There were hundreds of publications on this in 2020. Have a look at one of the many review papers, e.g. here is one with a meta-analysis [1].

Line 44: What’s the application, pls fix the end of the sentence.

Line 77: That’s a self-referencing definition. Please try to explain on a bit higher level (i.e., what’s the purpose of the code?) The explanation below is good, but this line is confusing.

L78: typo: windows

L187: bad cross-ref

L188: space missing after equation\\nL210: 212 what? apples? :D

Fig 4: Do you have permission from the copyright holders to print these images? Please doublecheck and cite.

L214: I have troubles understanding why this is imbalanced. The CXR dataset has 212 COVID-19 samples and 212 non-COVID samples (from the 8 diseases), no? Please clarify.

[1] Born, J, et al. "On the Role of Artificial Intelligence in Medical Imaging of COVID-19." medRxiv (2020).

[2] Li, Xin, Chengyin Li, and Dongxiao Zhu. "COVID-MobileXpert: On-device COVID-19 patient triage and follow-up using chest X-rays." 2020 IEEE International Conference on Bioinformatics and Biomedicine (BIBM). IEEE, 2020.

6. PLOS authors have the option to publish the peer review history of their article (what does this mean?). If published, this will include your full peer review and any attached files.

Reviewer #1: No

Reviewer #2: No

Reviewer #3: No

---

## [Author Response · Author response to Decision Letter 0]

30 Mar 2021

PLOS ONE

REPLY TO COMMENTS

Ref. No.: PONE-D-21-04444

Paper Title: COVID-19 Diagnosis from CT Scans and Chest X-ray Images using Low-cost Raspberry Pi.

Dear Editors and Reviewers,

Thank you for your valuable comments and feedback on our paper, which helped us improve its presentation and quality. We have carefully addressed all of your comments in the revised manuscript. We hope that you will be satisfied with the response provided by us.

Sincerely,

The authors.

Reviewer #1 Comments

Comment: 

far as one can see, the experimental work seems to have been carried out properly and the results are carefully presented, both in tables and in graphs, to compare this and earlier methods. However, the paper has following concern:

Response: 

Thank you for this valuable comment.

Comment 1: 

I dont understand exactely, why authors are fousing on Linux-embedded system, i.e., Raspberry Pi, to visually detect COVID-19 in chest X-ray and CT scans. Irrespective of platform, ultimately we can think of prediction accuracy with platform independent for better prospective.

Response 1: 

The authors are thankful to the reviewer for pointing out this issue. The proposed system is suitable for any computing device with the required environments (i.e., C++ and Python). The authors focused on validating their method on a Raspberry Pi embedded system module because Raspberry Pi modules are cheap and portable embedded systems; thus, developing countries can utilize the proposed work in remote areas to detect COVID-19 low budget. Also, the prediction accuracy is the same across different platforms. In other words, running the proposed machine learning model on any computing device/embedded system, the obtained accuracy will be the same. The main challenge was to propose a very lightweight (i.e., the proposed classification model is 3 MB) and efficient prediction model to fit embedded systems. We outlined this issue on page 2, lines 45-48.

Comment 2: 

Can the authors show the impacts of various proposed features to establish the proposal?

Response 2: 

The authors are thankful to the reviewer for pointing out this issue. This paper's main proposed feature is to extract the features from an X-ray or CT scan image and then classify this image under the limited memory space condition. Unlike deep learning methods, the proposed system is deployed using only 3MB, which is far less than the state-of-the-art methods requiring 100s of MB. Thus, the proposed work's main impact is that it becomes possible to predict COVI-19 cases from CT and X-ray images on an embedded system with minimal memory.

In the revised version of the manuscript, we added the following sentence, on page 2, lines 51-53. 

"The main impact of the proposed classifier model that it becomes possible to obtain a high level of accuracy rates under a limited memory condition predicting COVI-19 cases from CT and X-ray images."

Comment 3: 

The training conditions when comparing with other methods should be discussed more specifically. The paper lacks this part.

Response 3: 

The authors apologize for such inconvenience. In the revised manuscript, we discussed this issue in Section 5.1, on page 11, lines 297-302.

Comment 4:

 Do the authors take different combinations between training and testing samples in experimentations to show the merit of the proposed method? Proper justification should be added for better clarity.

Response 4: 

The authors are thankful to the reviewer for pointing out this issue. Yes, the authors utilized the cross-validation technique, where the proposed method is trained on five different combinations of datasets. This issue is discussed in Section 5.1 in the revised manuscript, on page 11, lines 297-302.

Comment 5: 

The title of the paper is very specific. I'll suggest modifying it to make it more generic using the term "directional codes".

Response 5: 

The authors are thankful to the reviewer for pointing out this issue. We prefer to use the same title. The authors emphasize using the Raspberry Pi model due to its widespread usage at low-cost. Thus, we believe mentioning the term "Raspberry Pi" in the title should attract readership and ease the task of reproducing the proposed experiments.

Comment 6: 

The title of the paper is very specific. I'll suggest modifying it to make it more generic using the term "directional codes".

Response 6: 

The authors are thankful to the reviewer for pointing out this issue. We prefer to use the same title. The authors emphasize using the Raspberry Pi model due to its widespread usage at low-cost. Thus, we believe mentioning the term "Raspberry Pi" in the title should attract readership and ease the task of reproducing the proposed experiments.

Comment 7: 

Whilst the English is generally quite good, there are quite a few minor grammatical errors, and a careful read through is needed to eliminate these.

Response 7: 

The authors are thankful to the reviewer for pointing out this issue. The paper is thoroughly revised, and all the typos and grammatical errors are addressed in the revised manuscript. We attached a certficate of reviewing the manuscript from the professional edidting service www.aje.com.

Comment 8: 

More experimental analysis need to be done for better claity to the end reader.

Response 8: 

The authors are thankful to the reviewer for pointing out this issue. In the revised manuscript, the authors extended the results (e.g., Figs. 10 and 11 were added to depict the proposed classifiers' confusion matrices). The obtained results are discussed in more detail; please refer to Section 5.3, on page 12, lines 330-339 and pages 15 and 16.

Comment 9: 

The objective and motivation need to address properly in abstract section. Current form of abstract will not accept by research community.

Response 9: 

The authors are thankful to the reviewer for pointing out this issue. The abstract section is rewritten as suggested. 

Comment 10: 

The major contribution of the paper should be heighlited immediately after literature kind review.

Response 10: 

The authors are thankful to the reviewer for pointing out this issue. As suggested, the revised manuscript includes the major contributions; please refer to page 2, lines 43-57. 

Comment: 

Hope these above points will be more beneficial to the author for improvement of the paper.

Response: 

The authors are thankful to the reviewer for pointing out this issue. All of the points were very helpful and helped the authors to enhance the paper quality.

Reviewer #2 Comments

Comment: 

This paper design a Raspberry Pi Linux embedded system for COVID-19 diagnosis from CT Scans and Chest X-ray Images. The cost of this system is the smallest among the deep learning-based models. My major concerns are list below:

Response: The authors are thankful for reviewer#2.

Comment 1: 

In the introduction section, should discuss some recent state-of-the-art models so that the readers can know more about the new techniques in this field. There are some literature that may be useful to further improve the quality of this section.

[ref1]Review of artificial intelligence techniques in imaging data acquisition, segmentation and diagnosis for covid-19, IEEE reviews in biomedical engineering, 2020;

[ref2]Inf-Net: Automatic COVID-19 Lung Infection Segmentation from CT Images，IEEE transactions on medical imaging, 2020;

[ref3]Joint prediction and time estimation of COVID-19 developing severe symptoms using chest CT scan, MIA, 2021;

[ref3]Severity assessment of COVID-19 using CT image features and laboratory indices, Physics in Medicine & Biology, 2020;

[ref4]Adaptive feature selection guided deep forest for covid-19 classification with chest ct, JBHI, 2020;

[ref5]JCS: An Explainable COVID-19 Diagnosis System by Joint Classification and Segmentation, TIP, 2021;

Response 1: 

The authors are thankful to the reviewer for pointing out these remarkable articles. As suggested, we added a new section (Section 3) for the literature review where we discussed the suggested related articles on pages 6 and 7. 

Comment 2: 

As seen in Fig.3, the proposed system belongs to the multi-modality system, thus, the authors should discuss more these works.

Response 2: 

The authors are thankful to the reviewer for pointing out this issue. We apologize for not clearly discuss this issue. The proposed system consists of two separate classifier models. Thus, there is no multi-modality in the proposed system. The user will input a chest X-ray image to the X-ray classifier or a CT scan image to the CT image classifier. Both models are designed the same way but separately. We have modified Fig. 3 to illustrate this idea. Besides, we mentioned that the two models are separate in the list of contributions in the Introduction Section, on page 2, lines 46-48 and on page 10, lines 283-284. 

Reviewer #3' Comments

Comment: 

This work presents a novel approach for rapid, on-device COVID-19 detection using Raspberry Pi. Despite the plethora of works on this topic, this one clearly stands-out due to the low computational requirements and the Raspberry Pi deployment. The experimental section is a bit short, but the results are convincing. I recommend acceptance, although a few things should be addressed:

Response: 

The authors are thankful for reviewer#3.

Comment 1: 

Is this the first scientific report of using Raspberry Pi to diagnose COVID from medical images? If yes, please state so and if not, please cite relevant work. I quickly searched but couldn't find anything very similar. I also recommend discussing the work by [2] since it also deals with on-device inference.

Response 1: 

The authors are thankful to the reviewer for pointing out this issue. Yes, the proposed work is the first work to use Raspberry Pi to diagnose COVID-19. As suggested, we stated that in both the Introduction and Conclusion sections. Besides, we discussed ref [2] in the revised manuscript on page 2, lines 43-48, and on page 12, lines 343, respectively. Besides, we discussed Ref [2] on page 6, lines 165-170.

Comment 2: 

Can you please comment on the overall time and space complexity? The elaboration about complexity in MFrLFMs is great (although there, at least a comment on space complexity would also be useful). It would be great to see a similar elaboration on the LBP method? I think if the entire pipeline (Fig 3) can be expressed in terms of computational complexity in O-notation, this could be a major finding and contribution that might even be worth mentioning in the abstract. Also the time complexity could be briefly compared with a standard MLP/CNN to solidify how important this contribution is.

Response 2: 

The authors are thankful to the reviewer for pointing out this issue. As suggested, the time and space complexities of the proposed work are discussed in detail a the end of Section 4.2, on pages 9-10, lines 238-260. Fig. 3 has two main compute-intensive tasks, which are local and global feature extraction. Thus, the time complexity and space complexity of Fig. 3 can be reduced to the sum of these two tasks. 

Regarding the time complexity of the MLP/CNN, several factors control this process, including the number of layers, the number of neurons per layer. Besides, the model hyperparameters' value, such as the early stopping, can dramatically change the time complexity of the MLP/CNN model. Thus, it would be challenging to compute the existing MLP/CNN models' exact time complexity. For space complexity, Table 3 in the revised manuscript shows each model's space requirements, which reflects the space complexity of the proposed method and state-of-the-art methods.

Comment 3: 

You performed a binary classification. Although it's questionable how practically relevant this is, it's beyond the scope of this work. But could you please briefly comment whether/how this could be extended to a multi-class classification.

Response 3: 

The authors are thankful to the reviewer for pointing out this issue. In the revised manuscript, we discussed this issue in conclusion about future work, on page 16, lines 356-358.

Comment 4: 

Table 1: I understand that these results are based on three repeated runs. Could you please at least show the performance with one more digit precision and also indicate the standard deviation. Ideally, a cross validation should be performed to strengthen this finding.

Response 4: 

The authors are thankful to the reviewer for pointing out this issue. As suggested, the required standard deviation is included in Table I. Besides, the authors already performed cross-validation. While this is not mentioned in the initial submission, it is discussed in the revised manuscript.

Comment 5: 

Please have a look at what "data availability" for this journal means. I understand the imaging data is public, but the requirements write e.g.: "For example, in addition to summary statistics, the data points behind means, medians and variance measures should be available." I therefore had to click "No" at one of the questions.

Response 5: 

The authors are thankful to the reviewer for pointing out this issue. 

Comment 6: 

Line 7 - 32: Correct me if I'm wrong, but this seems to be a random excerpt of the myriad of publications about COVID-19 detection with DL. I think these lines can be removed and replaced by a few, generic sentences about the efforts in the field. There were hundreds of publications on this in 2020. Have a look at one of the many review papers, e.g. here is one with a meta-analysis [1].

Response 6: 

The authors are thankful to the reviewer for pointing out this issue. As suggested, the authors summarized the literature work in a few generic sentences. A new section (Section 3) of the literature review is added on pages 6 and 7.

Comment 7: 

Line 44: What's the application, pls fix the end of the sentence.

Line 77: That's a self-referencing definition. Please try to explain on a bit higher level (i.e., what's the purpose of the code?) The explanation below is good, but this line is confusing.

L78: typo: windows

L187: bad cross-ref

L188: space missing after equation\\n

L210: 212 what? apples? :D

Response 7: 

The authors are thankful to the reviewer for pointing out all of these typos. We apologize for these typos. All of the typos are corrected.

Comment 8: 

Fig 4: Do you have permission from the copyright holders to print these images? Please doublecheck and cite.

Response 8: 

The authors are thankful to the reviewer for pointing out this issue. We made sure that the dataset is public and cited the dataset in the figure caption.

Comment 9: 

 L214: I have troubles understanding why this is imbalanced. The CXR dataset has 212 COVID-19 samples and 212 non-COVID samples (from the 8 diseases), no? Please clarify.

Response 9: 

We apologize for not writing this part. The class of non-COVID19 contains 212*8 = 1,696 images. On the other hand, the COVID-19 class has 212 images only. Thus, the two classes are unbalanced. We rewrote the unbalanced dataset in L214 in a better manner on page 10-11, lines 287-293.

[1] Born, J, et al. "On the Role of Artificial Intelligence in Medical Imaging of COVID-19." medRxiv (2020).

[2] Li, Xin, Chengyin Li, and Dongxiao Zhu. "COVID-MobileXpert: On-device COVID-19 patient triage and follow-up using chest X-rays." 2020 IEEE International Conference on Bioinformatics and Biomedicine (BIBM). IEEE, 2020.

---

## [Decision Letter · Decision Letter 1]

13 Apr 2021

COVID-19 Diagnosis from CT Scans and Chest X-ray Images using Low-cost Raspberry Pi

PONE-D-21-04444R1

Dear Dr. Hosny,

We’re pleased to inform you that your manuscript has been judged scientifically suitable for publication and will be formally accepted for publication once it meets all outstanding technical requirements.

Kind regards,

Gulistan Raja

Academic Editor

PLOS ONE

Additional Editor Comments (optional):

Reviewers' comments:

Reviewer's Responses to Questions

**Comments to the Author**

1. If the authors have adequately addressed your comments raised in a previous round of review and you feel that this manuscript is now acceptable for publication, you may indicate that here to bypass the “Comments to the Author” section, enter your conflict of interest statement in the “Confidential to Editor” section, and submit your "Accept" recommendation.

Reviewer #2: All comments have been addressed

Reviewer #3: All comments have been addressed

2. Is the manuscript technically sound, and do the data support the conclusions?

Reviewer #2: Yes

Reviewer #3: Yes

3. Has the statistical analysis been performed appropriately and rigorously? 

Reviewer #2: Yes

Reviewer #3: Yes

4. Have the authors made all data underlying the findings in their manuscript fully available?

Reviewer #2: No

Reviewer #3: No

5. Is the manuscript presented in an intelligible fashion and written in standard English?

Reviewer #2: Yes

Reviewer #3: Yes

6. Review Comments to the Author

Reviewer #2: The authors have adequately addressed my concerns and I recommend accepting the new version of manuscript.

Reviewer #3: Thanks for addressing the concerns sufficiently. Congratulations to the interesting and valuable project.

Here are my last thoughts:

- Fig 6 - 11 are really big and quite similar. I strongly recommend aggregating all of them into a single figure with subfigures.

- More of a recommendation: Since this is the first report of a raspberry Pi system for COVID detection from CXR, I would emphasize this in the abstract also.

- the data underlying the findings has to be made available (see guidelines)

7. PLOS authors have the option to publish the peer review history of their article (what does this mean?). If published, this will include your full peer review and any attached files.

Reviewer #2: No

Reviewer #3: **Yes: **Jannis Born

---

## [Editor Report · Acceptance letter]

30 Apr 2021

PONE-D-21-04444R1 

COVID-19 Diagnosis from CT Scans and Chest X-ray Images using Low-cost Raspberry Pi 

Dear Dr. Hosny:

I'm pleased to inform you that your manuscript has been deemed suitable for publication in PLOS ONE. Congratulations! Your manuscript is now with our production department. 

Kind regards, 

on behalf of

Dr. Gulistan Raja 

Academic Editor

PLOS ONE